# Toxicology and Microbiota: How Do Pesticides Influence Gut Microbiota? A Review

**DOI:** 10.3390/ijerph18115510

**Published:** 2021-05-21

**Authors:** Federica Giambò, Michele Teodoro, Chiara Costa, Concettina Fenga

**Affiliations:** 1Department of Biomedical and Dental Sciences and Morphofunctional Imaging, Occupational Medicine Section, University of Messina, 98125 Messina, Italy; federicagiambo@gmail.com (F.G.); micteodoro@unime.it (M.T.); cfenga@unime.it (C.F.); 2Clinical and Experimental Medicine Department, University of Messina, 98125 Messina, Italy

**Keywords:** gut microbiota, microbial community, pesticides, occupational exposure, dysbiosis

## Abstract

In recent years, new targets have been included between the health outcomes induced by pesticide exposure. The gastrointestinal tract is a key physical and biological barrier and it represents a primary site of exposure to toxic agents. Recently, the intestinal microbiota has emerged as a notable factor regulating pesticides’ toxicity. However, the specific mechanisms related to this interaction are not well known. In this review, we discuss the influence of pesticide exposure on the gut microbiota, discussing the factors influencing gut microbial diversity, and we summarize the updated literature. In conclusion, more studies are needed to clarify the host–microbial relationship concerning pesticide exposure and to define new prevention interventions, such as the identification of biomarkers of mucosal barrier function.

## 1. Introduction

In recent years, the demand for food has risen significantly in relation to the world population’s increase. At present, different types of pesticides are widely used worldwide to obtain better quality agricultural products and increase crop yields, thus bringing significant economic benefits. Pesticides reach the soil, water and air, and non-target organisms, including humans [1]. As a result, health concerns about pesticide risks to animals have increased in recent years [2,3,4]. A growing number of studies have reported that pesticides are linked to various pathologies, including metabolic diseases (such as obesity and type 2 diabetes) [5], dysregulation of the immune system [6,7,8], neurotoxicity [9], endocrine alterations, reproductive disorders [10,11], and even tumours [12,13], whereas the gastrointestinal microbiota critically contributes to a variety of host metabolic and immune functions.

Many efforts, focusing on the mechanisms of oxidative stress and the role played by the genetic profile in terms of susceptibility, have been conducted [14,15,16]. Recently, different studies have demonstrated that gut microbiota (GM) can be affected and altered by different types of environmental pollutants, including some pesticides [17], therefore the role of the gut microbiota in pesticide-induced toxicity in non-target organisms is gaining increasing attention [18].

The GM refers to the complex set of gut-resident bacteria, fungi and viruses that have mutualistic relationships with their host. About 10–100 trillion symbiotic microbial cells live in the human intestine [19]; the intestinal microbiota’s main components are bacteria species taxonomically classified by genus, family, order, and phyla. [20]. The GM is complex and challenging to characterize. The intestinal bacterial community can be mainly divided into 2172 species isolated in human beings, classified into 12 different phyla, of which 93.5% belong to *Proteobacteria, Firmicutes, Actinobacteria* and *Bacteroidetes* [21]. The microbiota composition and diversity remain relatively stable throughout life, although some changes have been observed in infancy and old age [22]. Several scientific studies point out that maintaining a healthy GM plays a key role in human health. The intestinal microbiota acts as an interface between foods, allows the assimilation of nutrients, helps the digestion of fibres, participates in the synthesis of some vitamins and amino acids, and regulates the absorption of fatty acids, calcium and magnesium. Furthermore, the gut microbiota is indispensable for the development and maturation of the gastrointestinal tract [23,24], a protective barrier against pathogenic microorganisms and toxins. The loss of stability of a healthy microbiota leads to the transition from eubiosis to dysbiosis and, consequently, pathological conditions ranging from inflammatory damage to chronic degenerative diseases, including tumours and neurological disorders [25,26,27].

In vitro, in vivo and epidemiological studies have linked human exposure to pesticides to several chronic diseases. In vivo, these effects have been investigated primarily in model organisms representing humans (such as rats or mice).

The mechanisms of interaction between pesticides and GM have been hypothesized in other studies [28,29] and will therefore be described shortly in this manuscript. Regarding how pesticides can interact with GM, the main types of interactions are illustrated in Figure 1. The GM metabolizes multiple chemicals after intake. Mutually, pesticides can impair the GM activity and composition with adverse repercussions for the host. Though it is known that GM perturbations can affect the homeostatic equilibrium between favourable and hostile microorganisms, it is not still clear how the GM and pesticides interact and whether these interactions are relevant for human health. In fact, it is reasonable to affirm that not all statistically significant perturbations are also biologically relevant and vice versa [30].

In this review, we mainly discussed the mechanisms by which different types of pesticides induce changes in microbiota composition and function, leading to the alteration of the hosts homeostasis. We believe that the gut microbiota, an underestimated target, might play a significant role in pesticide-induced toxicity.

## 2. Materials and Methods

A literature search was performed on PubMed and Scopus databases up to January 2021. The following search terms were used: ‘microbiota’ AND ‘pesticides’ (all fields), and ‘gut microbiota’ AND ‘pesticides’ (all fields). Only articles published in English were included. Additionally, supplementary articles were identified among the references of the screened articles. The attention was focused on recent findings with particular regard towards occupational exposure. The search produced a total of 1000 records. After removing duplicates, the eligibility of all articles detected was evaluated based on titles and abstracts. A total of 117 articles were selected for this review.

## 3. Herbicides

### 3.1. Glyphosate

Glyphosate (GLY) is the most popular herbicide used to eradicate plants and grasses. The use of glyphosate has been linked to adverse effects in mammals through various mechanisms [31]. Among these, alteration of the microbiota is one of the mechanisms under investigation. It has been demonstrated that the abundances of dominant GM species are decreased in bees exposed to GLY at standard environmental concentrations [32]. Some authors have described sublethal outcomes of GLY on the honeybee microbiota; GLY exposure reduced the amount of *Proteobacteria* and raised *Firmicutes* [33]. Besides, it has been hypothesized that the decrease in the intestinal bacteria leads to an increase in opportunistic pathogens; this might explain the increase in mortality in GLY-exposed bees [34]. Other studies have demonstrated that the maternal behaviour of rats exposed to GLY was altered. Additionally, neuroplasticity and GM resulted in being differently modulated by glyphosate alone or in a formulation (Roundup). *Bacteroidetes* and *Firmicutes* abundances showed differences in the Roundup group compared to the control and GLY [35]. Neurobehavioral effects in mice were also investigated after acute, sub-chronic and chronic exposure to GLY-based herbicides. Besides the reduction in *Actinobacteria* (*Corynebacterium*), *Firmicutes* (*Lactobacillus*) and *Bacteroidetes*, the authors also described the development of anxiety- and depression-like symptoms in the exposed animals [36]. Furthermore, in rats, it has been demonstrated that GLY can alter the morphological structure of the villi in the duodenum and jejunum. Also, GLY exposure reduced the antioxidant mechanisms of cells interfering with inflammatory pathways, since the mRNA expression levels of several mediators were increased after exposure (IL-1β, IL-6, TNF-α, MAPK3, NF-κB, and Caspase-3). The specific effect on the microbiota revealed a significant reduction in the abundance of *Firmicutes* associated with an enrichment of pathogenic bacteria [37]. Recently, the abnormal composition of GM has been linked to patients with autism spectrum disorder (ASD), suggesting that herbicide exposure during pregnancy may lead to an increased risk of developing ASD. However, the mechanisms behind these disorders have not been clarified yet. Juvenile offspring of mice exposed to GLY showed abnormalities in the gut microbiota composition and short-chain fatty acids in faecal samples. Moreover, the authors suggested that GLY exposure could be involved in ASD-like behaviours through the increased activity of soluble epoxide hydrolase in the brain of the exposed mice [38]. Contrariwise, a recent in vitro study analysing the intestinal microbiota of pigs did not show evident effects on the taxonomic level at the concentration of GLY administered [39]. Some authors assessed the effects of GLY on the microbiota of mussel ‘*M. galloprovincialis*’, suggesting that the altered proliferation of bacterial species after exposure to GLY could cause microbiota dysbiosis, inducing the spread of opportunistic pathogens [40]. The GM of the rats was affected by GLY exposure, alone or in a formulation; an accumulation of shikimic acid and a raised level of metabolites were found in cecum but not in serum, suggesting an effect restricted only on the gut microbiota [41]. The action of the glyphosate-based herbicide (Roundup) on the immune system and gut microbiota of crab ‘*Eriocheir sinensis*’ led to a reduction in digestive enzyme activities and in an assortment of microbiota, but amplified the abundance of *Bacteroidetes* and *Proteobacteria* at the taxonomic level [42]. In vitro, bacterial growth cultured from marine turtles’ guts exposed to GLY was strongly repressed. This lesser growth could be linked to altered digestion and overall health weakening in the turtles [43]. In quail ‘Coturnix japonica’ GLY-based herbicide exposure brought about a major alteration in females more than male animals, decreasing catalase activity in the liver and altered the GM [44].

### 3.2. Triazine, 2,4-Dichlorophenoxyacetic Acid

It has been shown that antibiotic treatment can alter the bioavailability of triazine pesticides in rats by modulating hepatic enzymes gene expression and gut absorption-related proteins, suggesting a role of *Firmicutes* [45]. Wang et al. investigated atrazine (ATZ) resistance in the wasp. They demonstrated how the impaired microbiota after acute exposure to ATZ was inherited across generations, even once exposure ceased [46]. Conversely, on a frog model, the authors demonstrated that ATZ did not affect microbiota at the estimated environmental concentration [47]. Other authors described a loss of mutualistic microbial species in oysters exposed to ATZ and a consequent rise of pathogenic bacteria [48]. Another popular herbicide, 2,4-dichlorophenoxyacetic acid (2,4-D), has been investigated in mice models. Animals were exposed to a sub-chronic low dose. A metagenomic and metabolomic analysis highlighted significant alterations in several microbial pathways. The 2,4-D considerably influenced the gut microbial assortment with *Bacteroidetes*, *Chlorobi*, *Chloroflexi*, *Spirochaetes* and *Thermotogae* enrichment in exposed mice [49]. See Table A1.

## 4. Organophosphate Pesticides (OPPs)

### 4.1. Chlorpyrifos

Chlorpyrifos (CPF) represents one of the most used organophosphorus pesticides. Many recent studies have investigated the effect of this toxicant in animal models, mainly rats and mice. Chronic exposure seems not only to affect the gut but also influence serum gonadotropins levels (follicle-stimulating hormone, luteinizing hormone and testosterone) and raise proinflammatory cytokines (such as IL-6, monocyte chemoattractant protein-1, and TNF-α) in rats, suggesting a potential role in the development of infertility and colitis [50]. Rats born from an exposed mother to oral CPF administration were smaller and lighter than non-exposed. Moreover, gut histological structures and the microbial community were altered; intestinal villi were shorter and thinner, and probiotic bacteria resulted as being lower in the ileum, caecum and colon [24]. It has been hypothesized that CPF could modify the gut microbiota composition in a diet-specific manner [17], even if authors observed differences only at the taxonomic level of genus and species but not at phylum [51]. Long-term exposure to CPF in mice could raise gut absorbency, reducing the gene expression of tight junction proteins such as occludin, claudin 1, and ZO-1, both in ileum and colon. It was observed that exposed mice gut microbiota showed an increment in *Proteobacteria* phylum and a decrement in *Bacteroidetes* phylum, suggesting that these phyla are mainly affected by CPF exposure [52]. Still, long-term exposure seems to amplify spontaneous vertical activity and female activity after acute stress in Wistar rats [53]. The host’s genetic profile can change the gut microbial response to CPF exposure. Guardia-Escote et al. orally exposed mice to CPF apoE3 and apoE4, observing differences between the two genotypes at different taxonomic levels; the phylum *Verrucomicrobia* resulted as highly represented only in apoE4 mice [54]. Subsequently to a daily dose of 1 mg/kg for 30 days in mice, besides altered gut microbiota, CPF also induced the alteration of urine metabolites, amino acids and short-chain fatty acids (SCFAs). As a result, the GM abundance of *Firmicutes* resulted as weakened, unlike *Bacteroidetes*, which resulted as higher [55]. Gut microbiota is supposed to be responsible for pesticide resistance [56]. A model organism silkworm, ‘*Bombyx mori*’, was fed with different antibiotics to induce gut dysbiosis and then exposed to CPF. It was observed that the antibiotic-fed silkworm larvae showed a significant amount of *Firmicutes* and *Actinobacteria* but a minor amount of *Proteobacteria*. Additionally, the antibiotic-treated silkworms seemed to be more vulnerable to CPF, supporting the hypothesis that the gut microbiota plays a crucial role in pesticide resistance [57]. Though, other authors did not demonstrate a shift in a mosquito gut microbial community, despite the pesticide-degrading microbiota resulting enriched [58]. In a *Drosophila melanogaster* model, *Lactobacillus plantarum* (*Firmicutes*) showed a greater capacity compared to *Acetobacter indonesiensis* (*Proteobacteria*), in the metabolization of CPF to the more effective metabolite chlorpyrifos oxon (CPO) [59].

Other authors highlighted CPF effects on oxidative stress and gut microbiota dysbiosis in zebrafish. They found an abundance of *Proteobacteria* with an alteration of *α-Proteobacteria*, *β*-*Proteobacteria,* and *γ*-*Proteobacteria* in CPF-treated zebrafishes compared to the control [60]. Still, through the use of an in vitro culture system, the gut microbiota of a marine benthic polychaete, ‘*Nereis succinea*’, showed fundamental metabolizing activity towards two representative organophosphates, CPF and malathion [61]. The impact of CPF on the gut microbiota has also been studied in vitro using the Simulator of the Human Intestinal Microbial Ecosystem (SHIME^®^) and Caco-2/TC7 cells of the intestinal mucosa. After 3.5 mg/day CPF exposure, *Lactobacillus* (*Firmicutes*) and *Bifidobacterium* (*Actinobacteria*) concentrations resulted as being strongly decreased [62]. A low-dose exposure (1 mg/day) after 15 and 30 days increased the *Enterobacteria* spp. *(Proteobacteria)*, *Bacteroides* spp. (*Bacteroidetes*) and *Clostridia* (*Firmicutes*) counts but decreased the *bifidobacterial (Actinobacteria)* counts, reducing fermentative efficacy [63]. Similar results were obtained from other authors who aimed to assess the effect of CPF chronic low-dose exposure (1 mg for 30 days), both on SHIME and rats. They observed an increment in the entire bacterial flora, primarily caused by *Enterococcus* spp. (*Firmicutes*) and *Bacteroides* spp. (*Bacteroidetes*), but a reduction in *bifidobacterial* (*Actinobacteria*) and *lactobacilli* (*Firmicutes*) [64].

### 4.2. Trichlorfon, Diazinon, Monocrotophos, Dichlorvos, Phoxim

The effects of trichlorfon administered to common carp at different concentrations (0, 0.1, 0.5 and 1.0 mg/L) were evaluated by Chang et al. Gut histological changes resulted in a reduced size of intestinal villus and an altered expression of tight junction proteins, such as claudin-2, occludin and ZO-1. In addition, many antioxidant enzymes were decreased and proinflammatory cytokines upregulated after treatment. The most represented phyla were *Fusobacteria*, *Proteobacteria* and *Bacteroidetes*. Trichlorfon administration led to a reduction in *Fusobacteria* and an increment in *Bacteroidetes* [65]. Inversely, diethyl phosphate exposure did not show proinflammatory effects in exposed rats. Opportunistic pathogens were found significantly enriched, but IL-6 was reduced [66]. The effects of another organophosphorus pesticide, diazinon, were evaluated in crucian carp. Diazinon exposure increased the counts of *Bacteroidetes, Fusobacteria* and *Firmicutes*, but not the abundance of *Proteobacteria.* After 14 days of exposure, also *Patescibacteria* were found to be incremented [67]. Low monocrotophos exposure led to a depletion of the bacterial and fungal populations in the earthworm gut (*Lampito mauritii*). After 15 days of exposure, the bacteria and fungi showed a minor increment, but on day 30, the total counts were lower than those of the control earthworms [68]. It has been hypothesized that gut microbiota could modulate resistance to dichlorvos in a species of beetles, *Callosobruchus maculatus* [69]; a similar mechanism has been suggested to explain the resistance of some strains of *Anopheles* mosquitoes to organophosphate fenitrothion [70]. Phoxim exposure in silkworms showed immunosuppressive effects and increased bacterial abundance, particularly in two genera, such as *Methylobacterium* and *Aurantimonadaceae* (*Proteobacteria*) [71]. See Table A2.

## 5. Organochlorine Pesticides (OCPs)

### p, p’-dichlorodiphenyldichloroethylene (p, p’-DDE), Dieldrin, Endosulfan, Indoxacarb, Hexachlorocyclohexane, Malachite Green

Dichlorodiphenyldichloroethylene (DDE) is the most prevalent metabolite of dichlorodiphenyltrichloroethane (DDT), a common organochloride pesticide banned several decades ago. Among all the adverse effects of organochlorides, the GM is also a target of these chemicals. In mice exposed for 8 weeks to DDE, alterations of gut microbiota composition resulted in an increment in *Bacteroidetes*, and a reduction in *Proteobacteria*, *Deferribacteres* and *Cyanobacteria* [72]. Contrariwise, in rats exposed to DDE, gut dysbiosis resulted in a reduction in *Bacteroidetes* and *Proteobacteria* accompanied by increased *Firmicutes* and *Tenericutes*, with a substantial rise in the *Firmicutes*-to-*Bacteroidetes* ratio [73]. Other authors investigated the effects of both DDE and β-hexachlorocyclohexane (β-HCH), an active metabolite of hexachlorocyclohexane (HCH), on mice, observing changes in hepatic and bile acid metabolism besides the aforementioned effects on the microbiota composition [74]. HCH, being a persistent organic pollutant, can alter the microbial composition in the human colostrum and, consequently, the colonization of the infant gut [75]. These results were confirmed by another study that investigated whether environmental pollutants, including organochlorine, in breastmilk could impact the homeostasis of the infant gut microbiota at one month, revealing differences in the abundance of some strains of *Firmicutes* [76]. Zebrafish fed with dieldrin did not show histological modifications of the gastrointestinal tract or body weight changes after four months of exposure. However, *Firmicutes* resulted as less abundant [77].

Malachite green (MG) (an organic chloride salt used as a fungicide in the aquaculture industry) has been shown to influence not only gut microbiota but also immune system homeostasis in goldfish. The altered gut microbiota was linked to opportunistic bacteria overgrowth and thus to potential infections [78]. Other authors observed a diet-specific effect of endosulfan sulphate, a major metabolite of the organochlorine insecticide endosulfan, in mice. The abundance of most the represented phyla (such as *Actinobacteria*, *Proteobacteria*, *Bacteroidetes* and *Firmicutes*) was different in the mice fed with a high-fat diet from low-fat diet fed mice [79]. The GM’s role against insecticide resistance has been demonstrated in the German cockroach. Indeed, using antibiotic treatments, the susceptibility to indoxacarb insecticide was incremented. The main phyla composing microbiota were *Proteobacteria*, *Bacteroidetes*, *Firmicutes* and *Fusobacteria*, and via faecal transplant, indoxacarb-resistant could be transferred to susceptible strains [80]. See Table A3.

## 6. Insecticides

### 6.1. Neonicotinoid, Imidacloprid

The toxicity of imidacloprid (IMI), a neonicotinoid insecticide, has received growing attention in the last years. In a *Drosophila melanogaster* model, IMI led to microbial dysregulation consisting of an increment in *Lactobacillus* spp. *(Firmicutes)* and *Acetobacter* spp. *(Proteobacteria)* [81]. Bees can be an indirect target of neonicotinoid insecticides such as IMI, thiacloprid, nitenpyram, amitraz and dimethoate. Several studies suggested that long-term exposure can lead to impairment of the gut microbial in bees with a reduced abundance of total bacterial flora, mainly *Firmicutes* [82,83,84,85,86]. In pregnant mice, nitenpyram also altered lipid metabolism and decreased the abundance of *Proteobacteria*, specifically of the Desulfovibrionaceae family [87], and IMI was shown to reduce liver weights and alter the histological structure of hepatic tissue, disrupting the gut microbial composition with consequently impaired barrier function [88]. Hong et al. evaluated the effects of IMI in crab, ‘*Erocheir sinensis*’. On the one hand, *Bacteroidetes* abundance was increased in a dose-dependent manner and reached almost three times as much as the control. On the other hand, the relative abundance of *Proteobacteria* diminished [89]. Onaru et al. described the effects of clothianidin on rats. The animals showed a loss of body weight and significant alterations in the gut microbiota, principally *Firmicutes* was strongly decreased, inversely *Bacteroidetes* was augmented and the dominance of *Proteobacteria* and *Actinobacteria* was inverted [90].

### 6.2. Pyrethroid

Among all pyrethroids, permethrin (PEM) is one of the most commonly used for pest control. In low-dose exposed rats, PEM decreased the amount of three microbial species belonging to the *Bacteroidetes phylum* (*Bacteroides*, *Prevotella* and *Porphyromonas*). Moreover, it showed antibacterial properties against beneficial bacteria such as *Bifidobacterium* (*Actinobacteria*) and *Lactobacillus* (*Firmicutes*) [91]. The widespread use of pyrethroids, such as alpha-cypermethrin, deltamethrin and PEM, in the ecosystem could modify the bacterial composition in mosquitoes, promoting insecticide-metabolizing microbiota, thus supporting insecticide resistance [92]. Further, methoprene, a biochemical insecticide, was capable of affecting the microbiota in larval mosquitoes, arising the abundances of Proteobacteria, and precisely the Gammaproteobacteria class and Enterobacteriaceae family [93]. In vitro analysis of intestinal epithelial cells confirms that gut microbiota activity can be disrupted by deltamethrin [18]. The gut microbial community in wasps has been suggested to determine animals’ susceptibility to insecticides such as abamectin [94]. Rouzé et al. investigated the effect of four pesticides (IMI, coumaphos, fipronil and thiamethoxam) on the GM of honeybees, highlighting how the microbial community’s composition differed in summer and winter bees. Sublethal doses of the compounds mentioned above showed similar effects and these were more substantial in the winter bees [95]. Other authors focused on the toxicity of aldicarb, a carbamate insecticide, on mice models, demonstrating through a metabolomic approach how this chemical can disrupt the gut microbiota leading to alteration in lipid profile, oxidative stress induction, DNA damage and susceptibility to bacterial pathogenicity [96]. See Table A4.

## 7. Fungicides

Fungicides aim to guard crops against parasitic fungi or spores. One of the most common fungicides is benzimidazole carbendazim (CBZ). It has been demonstrated that a 14-week exposure in mice induced changes in gut microbiota composition and an increment in body weight. Specifically, the amounts of *Bacteroidetes* and *Verrucomicrobia* were seriously lower, *Actinobacteria* increased significantly, while *Firmicutes* and *Proteobacteria* had no remarkable modification [97]. An additional study conducted on mice demonstrated that the bioaccumulation of CBZ was lower in the liver and gut compared to the faeces, suggesting a strong interaction with the gut microbial community until excretion. The faecal analysis showed a strong decrement in *Bacteroidetes* after five days of exposure; conversely, the relative abundance of *Firmicutes*, *Proteobacteria* and *Actinobacteria* increased [98]. The abundance of *Firmicutes*, *Bacteroidetes*, *Actinobacteria*, *Proteobacteria* and *Verrucomicrobia* were decreased in the gut of zebrafish exposed to CBZ for 21 days [99]. Otherwise, difenoconazole led to an increment in *Firmicutes*, *Proteobacteria* (*Aeromonas* and *Enterobacteriaceae*) and *Bacteroidetes* (*Bacteroides*) [100]. Xu et al. evaluated the chronic toxicity induced by epoxiconazole in rats. After 90 days of exposure, the richness of *Firmicutes* diminished while *Bacteroidetes* and *Proteobacteria* grew [101]. The mice exposed to penconazole showed a reduced amount of *Proteobacteria*, while *Bacteroidetes*, *Cyanobacteria* and *Actinobacteria* significantly increased. Both enantiomers toxicity was assessed and revealed substantial alteration in the abundances of gut microbial communities [102]. The fungicide azoxystrobin has proved able to alter the GM in soil earthworms, ‘Enchytraeus crypticus’, specifically *Proteobacteria*, which significantly increased after the 2 and 5 mg/kg exposure [103]. After 15 weeks of exposure to 2.5 mg/kg imazalil (IMZ), the microbial communities in the caecum and faeces of mice were significantly altered at the phylum and genus levels, with a substantial decline in *Bacteroidetes*, *Firmicutes* and *Actinobacteria* [104]. Previously, the same authors evaluated IMZ toxicity on the zebrafish model, highlighting an increment in *Fusobacteria* and *Firmicutes* and a decrement in *Proteobacteria* and *Bacteroidetes* after exposure to 1 mg/L IMZ for 21 days [105]. Even propamocarb (PM) showed significant alteration in the GM of zebrafish after 7 days of exposure to 1 mg/L. At the phylum level, *Proteobacteria*, *Bacteroidetes* and *Firmicutes* were increased [106]. PM has been shown to be able to cause alterations in the GM of mice exposed to 3, 30, and 300 mg/L for 28 days. At the phylum level, the abundance of *Firmicutes*, *Proteobacteria*, *Saccharibacteria*, *Actinobacteria* and *Tenericutes* were reduced, while *Bacteroidetes*, *Acidobacteria*, *Chlorobacteria* and *Planctomycetes* showed a growth [107]. In chickens, thiram exposure was shown to significantly alter the composition of the gut microbiota, affecting the relative abundance of *Firmicutes* and *Proteobacteria*. In addition, liver histopathological structure changes were observed, accompanied by the increased blood lipid parameters [108]. Wang et al. investigated the toxicity of two fungicides, chlorothalonil and procymidone, in mice. Both of the compounds impaired intestinal barrier function linked to glucolipid metabolism. Moreover, the histopathological analysis showed liver damage. Lastly, gut microbiota in the colon contents was significantly affected, with an increment in the *Firmicutes*-to-*Bacteroides* ratio [109]. See Table A5.

## 8. Pesticide Mixtures

Lukowicz et al. aimed to assess the toxicity of a mixture of six pesticides (boscalid, captan, chlorpyrifos, thiophanate, thiacloprid, and ziram) in mice, suggesting that pesticide treatment could affect the regular gut microbiota composition and the host metabolic homeostasis [110]. The perinatal exposure to a low-dose pesticide cocktail made up of six pesticides (chlorpyrifos, thiacloprid, thiophanate, captan, ziram and boscalid) administered to mice resulted in alterations in the urinary and faecal metabolite due to the alterations in the gut microbial communities [111].

## 9. Future Perspectives

Until now, most of the studies evaluating the interaction between chemical agents and microbiota have limited themselves to observing qualitative or quantitative alterations in the composition of the GM. Even if more and more studies are going beyond this approach by introducing mechanistic hypotheses and evaluating them on experimental models, mostly in vitro, it is necessary to clarify these interactions further at the molecular level.

Moreover, the evaluation of GM-mediated effects is hard to define, since not all GM perturbations will develop a disorder in the host. Thus, biologically significant perturbation on humans should be identified. For this purpose, an integration of innovative biology techniques must be used to gain new details on the critical GM perturbations resulting in a disease. All this effort is needed to obtain useful tools for a valid risk assessment in populations exposed for environmental or occupational reasons, and possibly obtain the necessary elements for any modification of the regulatory policies and threshold limit value.

## 10. Conclusions

In the last years, a growing body of evidence has demonstrated the genotoxic and epigenetic effects of pesticides predisposing for different diseases, including tumours, and autoimmune and neurodegenerative disorders [112,113,114,115,116]. The data presented in this review further elucidate the pathogenetic mechanisms of pesticides mediated by gut microbiota alteration. In this context, the GM has been associated with developing several disorders. The relationship between xenobiotics and microbiota is dualistic. On the one hand, pesticides can disrupt the typical composition and functionality of this complex system, leading to significant metabolic imbalances, especially in glycolipid metabolism. On the other hand, the bacterial community reply to pesticides toxicity by promoting the growth of those bacterial strains that are most involved in the mechanisms of the detoxification of these chemical compounds. The lack of mechanistic knowledge on the pathways leading to the qualitative and quantitative GM perturbation observed by most literature reports limits the real-life applications of the research outcomes. The microbial community needs to be further investigated, particularly its long-term impact on host health, with a view of prevention and as a potential toxicological marker of pesticide exposure.

The Table A1, Table A2, Table A3, Table A4 and Table A5 summarize the main studies reviewed, according with the subdivision of the chapters.

## Figures and Tables

**Figure 1 ijerph-18-05510-f001:**
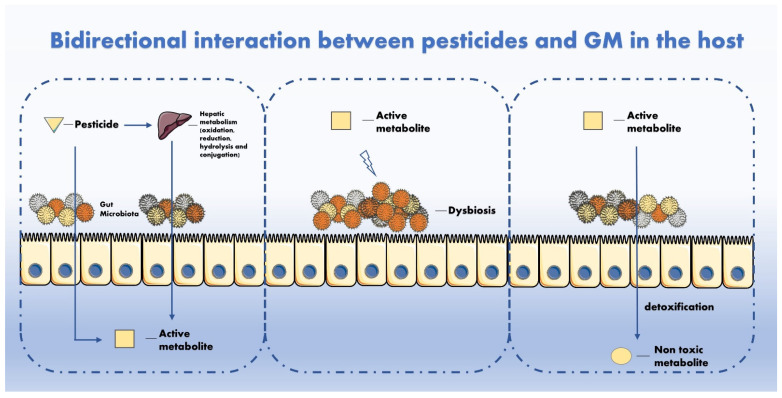
GM metabolizes multiple chemicals after intake. Mutually, pesticides can impair GM activity and composition with adverse repercussion for the host.

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
