# Peer review of "Toxicology and Microbiota: How Do Pesticides Influence Gut Microbiota? A Review"

_ijerph, 2021, doi:10.3390/ijerph18115510_

Round 1
Reviewer 1 Report
The manuscript entitled “Toxicology and Microbiota: How Do Pesticides Influence Gut Microbiota? A Review”provided a detailed literature of different types of pesticides inducing changes in microbiotal composition and function. This review clearly identified major achievements in the field in recent years, major research questions, or future research needs, which are key aspects. However, there are some concerns and points which must be improved. There are only one table and no figures to present the details of the use of pesticides, so it is hard to read the paper for IJERPH Audience. Authors are suggested to undergo the following correction as per the reviewer comments so as to improve the quality of manuscript.
-Abstracts needs to have more precision as in the current form it appears. In the abstract, please add an indication of the achievements from your study that is relevant to the journal scope. Please be concise - maximum 1-2 lines..
-The abstract of a good journal paper always ends outlining the benefits of the literature findings and recommendations as a way forward. The manuscript is missing such 1-2 lines in the abstract.
-Provide significant words which are more relevant to the work in logical sequence as ‘keywords’. Also use keywords which are not present in title.
-In introduction please improve the state of the art overview, to clearly show the progress beyond the state of the art. The lack of proper justification creates the wrong impression that the authors are unaware of the recent developments. Authors have done through literature survey and have presented the past works. But, what kind of innovation will be brought to the literature with this article? therefore, the state-of-art should be clearly specified in detail in the Introduction part. Hypothesis should be given. How this work is different from the available literature?
- What is the current level of understanding in relation to the potential effects of pesticide exposureconcerning the composition of gut microbiota? What are the knowledge gaps?. These should be included in the introduction section.
- Add 2-3 tables of the detailed process.
-It is also recommended to discuss and explain what should be the appropriate policies based on the findings of this study. Also, the literature should be further elaborated to show how they could be used for real applications.
-It is strongly recommended to add a new subsection, `future perspectives’ outlining the challenges in the current research and recommendations, before the conclusion.
-Conclusions: Conclusions are mainly based on suppositions and not on the literature evidence. All conclusions must be convincing statements on what was found to be novel, impactful based on strong support of the data/results/discussion. Limitations in the suggested approach should be discussed in the conclusions section, pls. conclude with more focus on the major outcomes of the paper.
Author Response
We are grateful to the Referees for their punctual and useful revision of this manuscript. We tried to respond to all the points raised as follows. Hoping to have fulfilled their requests.
Reviewer 1
- There are only one table and no figures to present the details of the use of pesticides, so it is hard to read the paper for IJERPH Audience.
R: Table 1 has been split into multiple tables with additional information; moreover, a figure has been added.
- Abstracts needs to have more precision as in the current form it appears. In the abstract, please add an indication of the achievements from your study that is relevant to the journal scope. Please be concise - maximum 1-2 lines.
- The abstract of a good journal paper always ends outlining the benefits of the literature findings and recommendations as a way forward. The manuscript is missing such 1-2 lines in the abstract.
R: Abstract has been revised as suggested.
- Provide significant words which are more relevant to the work in logical sequence as ‘keywords’. Also use keywords which are not present in title.
R: Keywords have been modified as requested.
- In introduction please improve the state of the art overview, to clearly show the progress beyond the state of the art. The lack of proper justification creates the wrong impression that the authors are unaware of the recent developments. Authors have done through literature survey and have presented the past works. But, what kind of innovation will be brought to the literature with this article? therefore, the state-of-art should be clearly specified in detail in the Introduction part. Hypothesis should be given. How this work is different from the available literature?
- What is the current level of understanding in relation to the potential effects of pesticide exposure concerning the composition of gut microbiota? What are the knowledge gaps? These should be included in the introduction section.
R: Thanks for these useful suggestions. Introduction section has been extensively revised and necessary corrections have been made. In particular, the state-of-art has been better detailed along with literature gap.
- Add 2-3 tables of the detailed process
R: Table 1 has been split into multiple tables with supplemental data and a figure illustrating mechanistic hypotheses of interaction between GM and pesticides has also been added.
- It is also recommended to discuss and explain what should be the appropriate policies based on the findings of this study. Also, the literature should be further elaborated to show how they could be used for real applications.
- It is strongly recommended to add a new subsection, `future perspectives’ outlining the challenges in the current research and recommendations, before the conclusion.
R: Thanks for advice. A new subsection has been added. Indication of specific policies regarding pesticides exposure is beyond the purposes of this review. However, we suggested as future perspectives a route leading from basic research to real life application, passing through risk assessment.
- Conclusions: Conclusions are mainly based on suppositions and not on the literature evidence. All conclusions must be convincing statements on what was found to be novel, impactful based on strong support of the data/results/discussion. Limitations in the suggested approach should be discussed in the conclusions section, pls. conclude with more focus on the major outcomes of the paper.
R: Conclusion and limitations have been revised according to suggestions.
Reviewer 2 Report
The manuscript entitled “Toxicology and microbiota: how do pesticides influence gut microbiota? A review” have analyzed 115 publications described „gut microbiota” and „pesticides”. I think the study is interesting, because describes important aspects related to the toxicity of pesticides and their impact on gut microbiota. However, the manuscript should be corrected.
Minor issues:
In my opinion Authors should insecticide DDT activity and its impact described too.
Table 1 - should be divided, e.g. according to "pesticide type/chemical structure" or "gut microbiota perturbation". In this form Table 1 is difficult to analyze
In Table 1 is information described "malachite green", cited reference is not properly, please correct
Author Response
We are grateful to the Referees for their punctual and useful revision of this manuscript. We tried to respond to all the points raised as follows. Hoping to have fulfilled their requests.
Reviewer 2
- Minor issues: In my opinion Authors should insecticide DDT activity and its impact described too.
R: DDT impact has not been described in detail as it is been banned from decades in several countries.
- Table 1 - should be divided, e.g. according to "pesticide type/chemical structure" or "gut microbiota perturbation". In this form Table 1 is difficult to analyze
R: Table 1 has been extensively revised.
- In Table 1 is information described "malachite green", cited reference is not properly, please correct.
R: In the reference list an author with the same surname contributed with a work published in the same year; we understand that this may have been misleading for the reviewer (see ref. 50 and 79).
Reviewer 3 Report
In this manuscript the authors performed a revision about the potential effects of pesticide exposure on gut microbiota composition. To achieve their aims they performed a search on Scopus and PubMed using several words selected by the authors on 2021.
In my opinion this theme of research is actual and important since microbiota changes and modifications are responsible for several health changes.
This manuscript can be accepted for publication, but it needs several corrections and improvements:
- The authors should improve the abstract quality. They should include a paragraph with the methodology used to write this revision and the main findings of this review;
-
I don't agree at all with the titles used for each topic covered in this review. Authors start by saying:
3. Effects of Herbicides on the Gut MicrobiotaThen next the topic is 4. Organophosphate Pesticides (OPPs). But aren't they talking about the effect on the gut microbiota? Then the authors go back to the topic 6. Effect of Insecticide on the Gut Microbiota. Authors should standardize all topics.
- Table 1: authors should better explain what in vitro tests were done to study microbiota: Were used cell lines? Bacteria? What was used in these in vitro studies?
- Table 1: Authors should correct the authors citation:
Motta et al (00), 2020. What is 00?
Some of the papers cited in table 1, make no reference to the disturbance caused in the microbial. This should be clarified by the authors.
-
Authors should check the references, as not all are formatted the same way.
-
After this review which future works should be done?
What impact this review may have on the development of new lines of research?
Author Response
We are grateful to the Referees for their punctual and useful revision of this manuscript. We tried to respond to all the points raised as follows. Hoping to have fulfilled their requests.
Reviewer 3
- The authors should improve the abstract quality. They should include a paragraph with the methodology used to write this revision and the main findings of this review;
R: Abstract section has been extensively revised.
- I don't agree at all with the titles used for each topic covered in this review. Authors start by saying: 3. Effects of Herbicides on the Gut Microbiota Then next the topic is 4. Organophosphate Pesticides (OPPs). But aren't they talking about the effect on the gut microbiota? Then the authors go back to the topic 6. Effect of Insecticide on the Gut Microbiota. Authors should standardize all topics.
R: As correctly suggested, all topic titles have been standardized.
- Table 1: authors should better explain what in vitro tests were done to study microbiota: Were used cell lines? Bacteria? What was used in these in vitro studies?
R: Experimental models has been specified for In vitro studies reported in table 1a.
- Table 1: Authors should correct the authors citation: Motta et al (00), 2020. What is 00? Some of the papers cited in table 1, make no reference to the disturbance caused in the microbial. This should be clarified by the authors.
R: This was a typo and it has been corrected.
- Authors should check the references, as not all are formatted the same way.
R: References list has been manually checked and revised.
- After this review which future works should be done? What impact this review may have on the development of new lines of research?
R: We suggested as future perspectives a route leading from basic research to real life application, passing through risk assessment.
Round 2
Reviewer 1 Report
The authors have addressed all the reviewer comments, therefore the manuscript may be accepted in the present form.